# Towards understanding the true loss surface of deep neural networks using random matrix theory and iterative spectral methods

## Abstract

The geometric properties of loss surfaces, such as the local flatness of a solution, are associated with generalization in deep learning. The Hessian is often used to understand these geometric properties. We investigate the differences between the eigenvalues of the neural network Hessian evaluated over the empirical dataset, the *Empirical Hessian*, and the eigenvalues of the Hessian under the data generating distribution, which we term the *True Hessian*. Under mild assumptions, we use random matrix theory to show that the True Hessian has eigenvalues of smaller absolute value than the Empirical Hessian. We support these results for different SGD schedules on both a 110-Layer ResNet and VGG-16. To perform these experiments we propose a framework for spectral visualization, based on GPU accelerated stochastic Lanczos quadrature. This approach is an order of magnitude faster than state-of-the-art methods for spectral visualization, and can be generically used to investigate the spectral properties of matrices in deep learning.

## 1 Introduction

The extraordinary success of deep learning in computer vision and natural language processing has been accompanied by an explosion of theoretical (Choromanska et al., 2015a;b; Pennington & Bahri, 2017) and empirical interest in their loss surfaces, typically through the study of the Hessian and its eigenspectrum (Ghorbani et al., 2019; Li et al., 2017; Sagun et al., 2016; 2017; Wu et al., 2017).

Exploratory work on the Hessian, and its evolution during training (e.g., Jastrzębski et al., 2018), attempts to understand why optimization procedures such as SGD can discover good solutions for training neural networks, given complex non-convex loss surfaces. For example, the ratio of the largest to smallest eigenvalues, known as the condition number, determines the convergence rate for first-order optimization methods on convex objectives (Nesterov, 2013). The presence of negative eigenvalues indicates non-convexity even at a local scale. Hessian analysis has also been a primary tool in further explaining the difference in generalization of solutions obtained, where under Bayesian complexity frameworks, flatter minima, which require less information to store, generalize better than sharp minima (Hochreiter & Schmidhuber, 1997). Further work has considered how large batch vs small batch stochastic gradient descent (SGD) alters the sharpness of solutions (Keskar et al., 2016), with smaller batches leading to convergence to flatter solutions, leading to better generalization. These geometrical insights have led to generalization procedures, such as taking the Cesàro mean of the weights along the SGD trajectory (Garipov et al., 2018; Izmailov et al., 2018), and algorithms that optimize the model to select for local flatness (Chaudhari et al., 2016). Flat regions of weight space are more robust under adversarial attack (Yao et al., 2018). Moreover, the Hessian defines the curvature of the posterior over weights in the Laplace approximation for Bayesian neural networks (MacKay, 1992; 2003), and thus crucially determines its performance.

In this paper we use random matrix theory to analyze the spectral differences between the *Empirical Hessian*, evaluated via a finite data sample (hence related to the empirical risk) and what we term the True Hessian, given under the expectation of the true data generating distribution.[1]

---

[1] We consider loss surfaces that correspond to risk surfaces in statistical learning theory terminology.

In particular, we show that the differences in extremal eigenvalues between the True Hessian and the Empirical Hessian depend on the ratio of model parameters to dataset size and the variance per element of the Hessian. Moreover, we show that that the Empirical Hessian spectrum, relative to that of the True Hessian, is *broadened*; i.e. the largest eigenvalues are larger and the smallest smaller. We support this theory with experiments on the CIFAR-10 and CIFAR-100 datasets for different learning rate schedules using a large modern neural network, the 110 Layer PreResNet.

It is not currently known if key results, such as (1) the flatness or sharpness of good and bad optima, (2) local non-convexity at the end of training, or (3) rank degeneracy hold for the True Hessian in the same way as for the Empirical Hessian. We hence provide an investigation of these foundational questions.

## 2 RELATED WORK, CONTRIBUTIONS AND PAPER STRUCTURE

Previous work has used random matrix theory to study the spectra of neural network Hessians under assumptions such as normality of inputs and weights, to show results such as the decreasing difference in loss value between the local and global minima (Choromanska et al., 2015a) and the fraction of negative eigenvalues at final points in training (Pennington & Bahri, 2017). To the authors knowledge no theoretical work has been done on the nature of the difference in spectra between the True Hessian and Empirical Hessian in machine learning. Previous empirical work on neural network loss surfaces (Ghorbani et al., 2019; Papyan, 2018; Sagun et al., 2017; 2016; Jastrzębski et al., 2018) has also exclusively focused on the Empirical Hessian or a sub-sample thereof.

The work closest in spirit to our work is the spiked covariance literature, which studies the problem of learning the true covariance matrix given by the generating distribution, from the noisy sample covariance matrix. This problem is studied in mathematics and physics (Baik & Silverstein, 2004; Bloemendal et al., 2016b;a) with applications leading to extensively improved results in both sparse Principal Component Analysis (Johnstone & Lu, 2004), portfolio theory (Laloux et al., 1999) and Bayesian covariance matrix spectrum estimation (Everson & Roberts, 2000). A key concept from this literature, is the ratio of parameters to data samples and its effect on the observed spectrum; this ratio is mentioned in (Pennington & Bahri, 2017), but it is not used to determine the perturbation between the Empirical Hessian and True Hessian. (Kunstner et al., 2019) consider circumstances under which the Empirical Fisher information matrix is a bad proxy to the True Fisher matrix, but do not use a random matrix theory to characterise the spectral perturbations between the two. We list our main contributions below.

- We introduce the concept of the True Hessian, discuss its importance and investigate it both theoretically and empirically.

- We use random matrix theory to analytically derive the eigenvalue perturbations between the True Hessian and the Empirical Hessian, showing the spectrum of the Empirical Hessian to be broadened.

- We visualize the True Hessian spectrum by combining a GPU accelerated stochastic Lanczos quadrature (Gardner et al., 2018) with data-augmentation. Our spectral visualization technique is an order of magnitude faster than recent iterative methods (Ghorbani et al., 2019; Papyan, 2018), requires one less hand-tuned hyper-parameter, and is consistent with the observed moment information. This procedure can be generically used to compute the spectra of large matrices (e.g., Hessians, Fisher information matrices) in deep learning.

## 3 PROBLEM FORMULATION

For an input $\boldsymbol{x} \in \mathbb{R}^{d_x}$ and output $\boldsymbol{y} \in \mathbb{R}^{d_y}$ we have a given prediction function $h(\cdot; \cdot) : \mathbb{R}^{d_x} \times \mathbb{R}^P \to \mathbb{R}^{d_y}$, we consider the family of prediction functions parameterised by a weight vector $\boldsymbol{w}$, i.e., $\mathcal{H} := \{h(\cdot; \boldsymbol{w}) : \boldsymbol{w} \in \mathbb{R}^P\}$ with a given loss $\ell(h(\boldsymbol{x}; \boldsymbol{w}), \boldsymbol{y}) : \mathbb{R}^{d_x} \times \mathbb{R}^{d_y} \to \mathbb{R}$. Ideally we would vary $\boldsymbol{w}$ such that we minimize the loss over our data generating distribution $\psi(\boldsymbol{x}, \boldsymbol{y})$, known as the *true risk*.

$$R_{true}(\boldsymbol{w}) = \int_{\mathbb{R}^{d_x} \times \mathbb{R}^{d_y}} \ell(h(\boldsymbol{x}; \boldsymbol{w}), \boldsymbol{y}) d\psi(\boldsymbol{x}, \boldsymbol{y}) = \mathbb{E}[\ell(h(\boldsymbol{x}; \boldsymbol{w}), \boldsymbol{y})] \tag{1}$$

with corresponding gradient $\boldsymbol{g}_{true}(\boldsymbol{w}) = \nabla R_{true}(\boldsymbol{w})$ and Hessian $\boldsymbol{H}_{true}(\boldsymbol{w}) = \nabla\nabla R_{true}(\boldsymbol{w})$. If the loss is a negative log likelihood, then the true risk is the expected negative log likelihood per data point under the data generating distribution. However, given a dataset of size $N$, we only have access to the *empirical* or full risk

$$R_{emp}(\boldsymbol{w}) = \frac{1}{N}\sum_{i=1}^{N}\ell(h(\boldsymbol{x}_i;\boldsymbol{w}),\boldsymbol{y}_i) \tag{2}$$

and the gradients $\boldsymbol{g}_{emp}(\boldsymbol{w})$ and Hessians $\boldsymbol{H}_{emp}(\boldsymbol{w})$ thereof. The Hessian describes the curvature at that point in weight space $\boldsymbol{w}$ and hence the risk surface can be studied through the Hessian. Weight vectors which achieve low values of equation 2 do not necessarily achieve low values of equation 1. Their difference is known as the *generalization gap*. We rewrite our Empirical Hessian as

$$\boldsymbol{H}_{emp}(\boldsymbol{w}) = \boldsymbol{H}_{true}(\boldsymbol{w}) + \boldsymbol{\varepsilon}(\boldsymbol{w}) \tag{3}$$

where[2] $\boldsymbol{\varepsilon}(\boldsymbol{w}) \equiv \boldsymbol{H}_{emp}(\boldsymbol{w}) - \boldsymbol{H}_{true}(\boldsymbol{w})$. A symmetric matrix with independent normally distributed elements of equal variance is known as the Gaussian Orthogonal Ensemble (GOE) and a matrix with independent non-Gaussian elements of equal variance is known as the Wigner ensemble.

### 3.1 FLATNESS AND GENERALIZATION WITH THE EMPIRICAL HESSIAN

By the spectral theorem, we can rewrite $\boldsymbol{H}_{emp}$ in terms of its eigenvalue, eigenvector pairs, $[\lambda_i, \boldsymbol{\phi}_i]$:

$$\boldsymbol{H}_{emp} = \sum_{i=1}^{P}\lambda_i\boldsymbol{\phi}_i\boldsymbol{\phi}_i^T. \tag{4}$$

The magnitude of the eigenvalues represent the magnitude of the curvature in the direction of the corresponding eigenvectors. Often the magnitude of the largest eigenvalue $\lambda_1$, or the Frobenius norm (given by the square root of the sum of all the eigenvalues squared $(\sum_{i=1}^{P}\lambda_i^2)^{1/2}$), is used to define the sharpness of an optimum (e.g., Jastrzębski et al., 2018; 2017). The normalized mean value of the spectrum, also known as the trace $P^{-1}\sum_i^P\lambda_i^2$, has also been used (Dinh et al., 2017). In all of these cases a larger value indicates greater sharpness. Other definitions of flatness have looked at counting the number of $0$ (Sagun et al., 2016), or close to $0$ (Chaudhari et al., 2016), eigenvalues.

It is often argued that flat solutions provide better generalization because they are more robust to shifts between the train and test loss surfaces (e.g., Keskar et al., 2016) that exist because of statistical differences between the training and test samples. It is then compelling to understand whether flat solutions associated with the *True Hessian* would also provide better generalization, since the True Hessian is formed from the full data generating distribution. If so, there may be other important reasons why flat solutions provide better generalization.

Reparametrization can give the appearance of flatness or sharpness from the perspective of the Hessian. However, it is common practice to hold the parametrization of the model *fixed* when comparing the Hessian at different parameter settings, or evaluated over different datasets.

## 4 ANALYZING THE TRUE RISK SURFACE USING RANDOM MATRIX THEORY

We show that the elementwise difference between the True and Empirical Hessian converges to a zero mean normal random variable, assuming a Lipshitz bounded gradient. Moreover, we derive a precise analytic relationship between the values of the extremal values for both the Empirical and True Hessians, showing the eigenvalues for the Empirical Hessian to be larger in magnitude. This result indicates that the true risk surface is flatter than its empirical counterpart.

### 4.1 REGULARITY CONDITIONS

We establish some weak regularity conditions on the loss function and data, under which the elements of $\boldsymbol{\varepsilon}(\boldsymbol{w})$ converge to zero mean normal random variables.

---

[2]For a fixed dataset, the perturbing matrix $\boldsymbol{\varepsilon}(\boldsymbol{w})$ can be seen as a fixed instance of a random variable.

**Lemma 1.** *For an L-Lipshitz-continuous-gradient and almost everywhere twice differentiable loss function $\ell(h(\boldsymbol{x}; \boldsymbol{w}), \boldsymbol{y})$, the True Hessian elements are strictly bounded in the range $-\sqrt{P}L \leq \boldsymbol{H}_{j,k} \leq \sqrt{P}L$.*

*Proof.* By the fundamental theorem of calculus and the definition of Lipshitz continuity $\lambda_{max} \leq L$

$$\text{Tr}(\boldsymbol{H}^2) = \sum_{i=1}^{P} \lambda_i^2 = \sum_{j,k=1}^{P} \boldsymbol{H}_{j,k}^2 = \boldsymbol{H}_{j=j',k=k'}^2 + \sum_{j \neq j', k \neq k'}^{P} \boldsymbol{H}_{j,k}^2$$

$$\boldsymbol{H}_{j=j',k=k'}^2 \leq \sum_{i=1}^{P} \lambda_i^2 \leq PL^2 \tag{5}$$

$$-\sqrt{P}L \leq \boldsymbol{H}_{j=j',k=k'} \leq \sqrt{P}L$$

$\square$

**Lemma 2.** *For unbiased independent samples drawn from the data generating distribution and an L-Lipshitz loss $\ell$ the difference between the True Hessian and Empirical Hessian converges element-wise to a zero mean, normal random variable with variance $\propto 1/N$.*

*Proof.* The difference between the Empirical and True Hessian $\boldsymbol{\varepsilon}(\boldsymbol{w})$ is given as

$$[\nabla\nabla R_{true}(\boldsymbol{w}) - \nabla\nabla R_{emp}(\boldsymbol{w})]_{jk} = \mathbb{E}\frac{\partial^2}{\partial \boldsymbol{w}_j \partial \boldsymbol{w}_k}\ell(h(\boldsymbol{x}; \boldsymbol{w}), \boldsymbol{y}) - \frac{1}{N}\sum_{i=1}^{N}\frac{\partial^2}{\partial \boldsymbol{w}_j \partial \boldsymbol{w}_k}\ell(h(\boldsymbol{x}_i; \boldsymbol{w}), \boldsymbol{y}_i) \tag{6}$$

By Lemma 1, the Hessian elements are bounded, hence the moments are bounded and using independence and the central limit theorem, equation 6 converges almost surely to a normal random variable $\mathbb{P}(\mu_{jk}, \sigma_{jk}^2/N)$.

$\square$

*Remark.* For finite $P$ and $N \to \infty$, i.e. $q = P/N \to 0$, $|\boldsymbol{\varepsilon}(\boldsymbol{w})| \to 0$ we recover the True Hessian. Similarly in this limit our empirical risk converges almost surely to our true risk, i.e. we eliminate the *generalization gap*. However, in deep learning typically the network size eclipses the dataset size by orders of magnitude.[3]

### 4.2 A RANDOM MATRIX THEORY APPROACH

In order to derive analytic results, we move to the large Dimension limit, where $P, N \to \infty$ but $P/N = q > 0$ and employ the machinery of random matrix theory to derive results for the perturbations on the eigenspectrum between the True Hessian and Empirical Hessian. This differs from the classical statistical regime where $q \to 0$. We primarily focus on the regime when $q \gg 1$. In order to make the analysis tractable, we introduce two further assumptions on the nature of the elements $\boldsymbol{\varepsilon}(\boldsymbol{w})$.

**Assumption 1.** *The elements of $\boldsymbol{\varepsilon}(\boldsymbol{w})$ are identically and independently Gaussian distributed $\mathcal{N}(0, \sigma_\epsilon^2)$ up to the Hermitian condition.*

**Assumption 2.** *$\boldsymbol{H}_{true}$ is of low rank $r \ll P$.*

Under assumption 1, $\boldsymbol{\varepsilon}(\boldsymbol{w})$ becomes a Gaussian Orthogonal Ensemble (GOE) and for the GOE we can prove the following Lemma 3. We discuss the necessity of the assumptions in Section 4.3.

**Lemma 3.** *The extremal eigenvalues $[\lambda_1', \lambda_P']$ of the matrix sum $\boldsymbol{A} + \boldsymbol{B}/\sqrt{P}$, where $\boldsymbol{A} \in \mathbb{R}^{P \times P}$ is a matrix of finite rank $r$ with extremal eigenvalues $[\lambda_1, \lambda_P]$ and $\boldsymbol{B} \in \mathbb{R}^{P \times P}$ is a GOE matrix with element variance $\sigma_\epsilon^2$ are given by*

$$\lambda_1' = \left\{ \begin{array}{ll} \lambda_1 + \frac{\sigma_\epsilon^2}{\lambda_1}, & \text{if } \lambda_1 > \sigma_\epsilon \\ 2\sigma_\epsilon, & \text{otherwise} \end{array} \right\}, \quad \lambda_P' = \left\{ \begin{array}{ll} \lambda_P + \frac{\sigma_\epsilon^2}{\lambda_n}, & \text{if } \lambda_n < -\sigma_\epsilon \\ -2\sigma_\epsilon, & \text{otherwise} \end{array} \right\} \tag{7}$$

---

[3]CIFAR datasets, which have $50,000$ examples, are routinely used to train networks with about 50 million parameters.

*Proof.* See Appendix. □

**Theorem 1.** *The extremal eigenvalues* $[\lambda'_1, \lambda'_P]$ *of the matrix sum* $\boldsymbol{H}_{emp}$, *where* $\lambda'_1 \geq \lambda'_2... \geq \lambda'_P$ *(such that assumptions 1 and 2 are satisfied), where* $H_{true}$ *has extremal eigenvalues* $[\lambda_1, \lambda_P]$, *are given by*

$$\lambda'_1 = \left\{ \begin{array}{ll} \lambda_1 + \frac{P}{N}\frac{\sigma_\epsilon^2}{\lambda_1}, & \text{if } \lambda_1 > \sqrt{\frac{P}{N}}\sigma_\epsilon \\ 2\sqrt{\frac{P}{N}}\sigma_\epsilon, & \text{otherwise} \end{array} \right\}, \lambda'_P = \left\{ \begin{array}{ll} \lambda_P + \frac{P}{N}\frac{\sigma_\epsilon^2}{\lambda_P}, & \text{if } \lambda_P < -\sqrt{\frac{P}{N}}\sigma_\epsilon \\ -2\sqrt{\frac{P}{N}}\sigma_\epsilon, & \text{otherwise} \end{array} \right\}. \quad (8)$$

*Proof.* The result follows directly from Lemmas 2 and 3. □

*Remark.* This result shows that the extremal eigenvalues of the Empirical Hessian are larger in magnitude than those of the True Hessian: Although we only state this result for the extremal eigenvalues, it holds for any number of well separated outlier eigenvalues. This spectral broadening effect has already been observed empirically when moving from (larger) training to the (smaller) test set (Papyan, 2018). Intuitively variance of the Empirical Hessian eigenvalues can be seen as the variance of the True Hessian eigenvalues plus the variance of the Hessian, as we show in Appendix F. We also note that if the condition $|\lambda_i| > \sqrt{P/N}\sigma_\epsilon$ is not met, the value of the positive and negative extremal eigenvalues are completely determined by the noise matrix.

### 4.3 NOTE ON GENERALIZING ASSUMPTIONS

For Assumption 1, we note that the results for the Wigner ensemble (of which the GOE is a special case) can be extended to non-identical element variances (Tao, 2012) and element dependence (Götze et al., 2012; Schenker & Schulz-Baldes, 2005). Hence, similarly to Pennington & Bahri (2017), under extended technical conditions we expect our results to hold more generally. Assumption 2 can be relaxed if the number of extremal eigenvalues is much smaller than $P$, and the bulk of $\boldsymbol{H}_{true}(\boldsymbol{w})$'s eigenspectra also follows a Wigner ensemble which is mutually free with that of $\boldsymbol{\varepsilon}(\boldsymbol{w})$. Furthermore, corrections to our results for finite $P$ scale as $P^{-1/4}$ for matrices with finite 4'th moments and $P^{-2/5}$ for all finite moments (Bai, 2008).

## 5 EFFICIENT COMPUTATION OF HESSIAN EIGENVALUES

In order to perform spectral analysis on the Hessian of typical neural networks, with tens of millions of parameters, we avoid the infeasible $\mathcal{O}(P^3)$ eigen-decomposition and use the stochastic Lanczos quadrature (SLQ) algorithm, in conjunction with GPU acceleration. Our procedure in this section is a general-purpose approach for efficiently computing the spectra of large matrices in deep learning.

### 5.1 STOCHASTIC LANCZOS QUADRATURE

The Lanczos algorithm (Meurant & Strakoš, 2006) is a power iteration algorithm variant which by enforcing orthogonality and storing the Krylov subspace, $\mathcal{K}_{m+1}(H, v) = \{v, Hv, .., H^m v\}$, optimally approximates the extremal and interior eigenvalues (known as Ritz values). It requires Hessian vector products, for which we use the Pearlmutter trick (Pearlmutter, 1994) with computational cost $\mathcal{O}(NP)$, where $N$ is the dataset size and $P$ is the number of parameters. Hence for $m$ steps the total computational complexity including re-orthogonalisation is $\mathcal{O}(NPm)$ and memory cost of $\mathcal{O}(Pm)$. In order to obtain accurate spectral density estimates we re-orthogonalise at every step (Meurant & Strakoš, 2006). We exploit the relationship between the Lanczos method and Gaussian quadrature, using random vectors to allow us to learn a discrete approximation of the spectral density. A quadrature rule is a relation of the form,

$$\int_a^b f(\lambda)d\mu(\lambda) = \sum_{j=1}^{M} \rho_j f(t_j) + R[f] \quad (9)$$

for a function $f$, such that its Riemann-Stieltjes integral and all the moments exist on the measure $d\mu(\lambda)$, on the interval $[a, b]$ and where $R[f]$ denotes the unknown remainder. The nodes $t_j$ of the Gauss quadrature rule are given by the Ritz values and the weights (or mass) $\rho_j$ by the squares of the

first elements of the normalized eigenvectors of the Lanczos tri-diagonal matrix (Golub & Meurant, 1994). For zero mean, unit variance random vectors, using the linearity of trace and expectation

$$\mathbb{E}_{\boldsymbol{v}}\text{Tr}(\boldsymbol{v}^T\boldsymbol{H}^m\boldsymbol{v}) = \text{Tr}\mathbb{E}_{\boldsymbol{v}}(\boldsymbol{v}\boldsymbol{v}^N\boldsymbol{H}^m) = \text{Tr}(\boldsymbol{H}^m) = \sum_{i=1}^{P}\lambda_i^m = P\int_{\lambda\in\mathcal{D}}\lambda^m d\mu(\lambda) \qquad (10)$$

hence the measure on the LHS of equation 9 corresponds to that of the underlying spectral density. The error between the expectation over the set of all zero mean, unit variance vectors $v$ and the Monte Carlo sum used in practice can be bounded (Hutchinson, 1990; Roosta-Khorasani & Ascher, 2015), but the bounds are too loose to be of any practical value (Granziol & Roberts, 2017). However, in the high dimensional regime $N \to \infty$, we expect the squared overlap of each random vector with an eigenvector of $\boldsymbol{H}$, $|\boldsymbol{v}^T\boldsymbol{\phi}_i|^2 \approx \frac{1}{P}\forall i$, with high probability (Cai et al., 2013). This result can be seen intuitively by looking at the normalized overlap squared between two random Rademacher vectors, where each element is drawn from the distribution $\pm 1$ with equal probability, which gives $1/P$. The Lanczos algorithm can also be easily parallelized, since it largely involves matrix multiplications — a major advantage for computing eigenvalues of large matrices in deep learning. We exploit GPU parallelization in performing Lanczos (Gardner et al., 2018), for significant acceleration.

## 5.2 WHY TO AVOID MULTIPLE RANDOM VECTORS AND KERNEL SMOOTHING

In recent work using SLQ for evaluating neural network spectra (Papyan, 2018; Ghorbani et al., 2019), the Hessian vector product is averaged over a set of random vectors, typically $\approx 10$. The resulting discrete moment matched approximation to the spectra in equation 9 is smoothed using a Gaussian kernel $\mathcal{N}(\lambda_i, \sigma^2)$. The use of multiple random vectors can be seen as reducing the variance of the vector overlap in a geometric equivalent of the central limit theorem. Whilst this may be desirable for general spectra, if we believe the spectra to be composed of outliers and a bulk as is empirically observed (Sagun et al., 2016; 2017), then the same self-averaging procedure happens with the bulk. We converge to the spectral outliers quickly using the Lanczos algorithm, as shown in the Appendix by the known convergence theorem 3. Hence all the spectral information we need can be gleaned from a single random vector. Furthermore, it has been proven that kernel smoothing biases the moment information obtained by the Lanczos method (Granziol et al., 2018). We hence avoid smoothing (and the problem of choosing $\sigma$), along with the $\approx 10\times$ increased computational cost, by plotting the discrete spectral density implied by equation 9 for a single random vector, in the form of a stem plot.

## 5.3 WEAKNESSES OF STOCHASTIC LANCZOS QUADRATURE

It is possible to determine what fraction of eigenvalues is near the origin by evaluating the weight of the Ritz value(s) closest to the origin. But due to the discrete nature of the Ritz values, which give the nodes of the associated moment-matched spectral density in equation 9, it is not possible to see exactly whether the Hessian of neural networks is rank degenerate. It is also not possible to exactly determine what fraction of eigenvalues are negative. Furthermore, smoothing the density and using quadrature has the problems discussed in Section 5.2. Instead, we determine the fraction of negative eigenvalues as the weight of negative Ritz values which are well separated from the smallest magnitude Ritz value(s), to avoid counting a split of the degenerate mass.

## 6 EXPERIMENTS

In order to test the validity of our theoretical results in Section 4.2, we control the parameter $q = P/N$ by applying data-augmentation and then running GPU powered Lanczos method (Gardner et al., 2018) on the augmented dataset at the final set of weights $\boldsymbol{w}_{final}$ at the end of the training procedure. We use random horizontal flips, $4 \times 4$ padding with zeros and random $32 \times 32$ crops. We neglect the dependence of the augmented dataset on the input dataset in our analysis, but we would expect the effective number of data points $N_{eff} < N$ due to degeneracy in the samples. For the Empirical Hessian, we use the full training and test dataset $N = 50,000$ with no augmentation. We use PyTorch (Paszke et al., 2017). We train a 110 layer pre-activated ResNet (He et al., 2016) neural network architecture on CIFAR-10 and CIFAR-100 using SGD with momentum set at $\rho = 0.9$ and data-augmentation. We include further plots and detail the training procedure for the VGG-16 (Simonyan & Zisserman, 2014) in Appendix I.

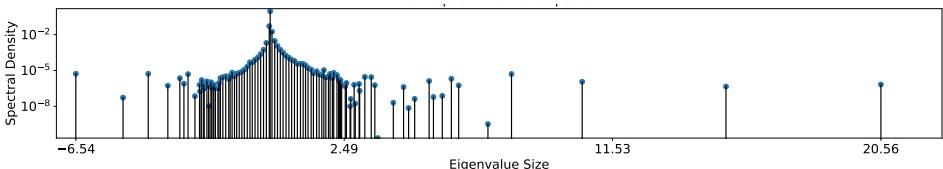

Figure 1: Empirical Hessian spectrum for CIFAR-100 SGD.

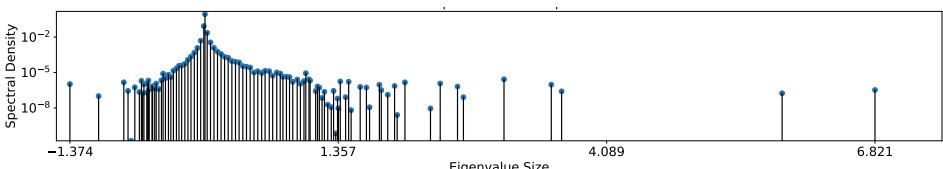

Figure 2: Empirical Hessian spectrum for CIFAR-10 SGD.

In order to investigate whether empirically observed phenomena for the Empirical Hessian such as sharp minima generalizing more poorly than their flatter counterparts also hold for the True Hessian, we run two slightly different SGD decayed learning rate schedules (equation 11) on a 110-Layer ResNet (Izmailov et al., 2018).

$$\alpha_t = \begin{cases} \alpha_0, & \text{if } \frac{t}{T} \leq 0.5 \\ \alpha_0[1 - \frac{(1-r)(\frac{t}{T}-0.5)}{0.4}] & \text{if } 0.5 < \frac{t}{T} \leq 0.9 \\ \alpha_0 r, & \text{otherwise} \end{cases} \tag{11}$$

For both schedules we use $r = 0.01$. For the *Normal* schedule we use $\alpha_0 = 0.1$ and $T = 225$ whereas for the *Overfit* schedule we use $\alpha_0 = 0.001$ and $T = 1000$.

We plot the training curves in the Appendices B, C and include the best training and test accuracies and losses, along with the extremal eigenvalues of the Empirical Hessian and Augmented Hessian in Table 1 for CIFAR-100 and Table 2 for CIFAR-10. The Normal schedule achieves a best test accuracy of [77.24/95.32] on CIFAR-100/CIFAR-10 respectively and best training accuracy of [99.85/99.98]. The Overfit schedule achieves a best test accuracy of [56.85/86.46] on CIFAR-100/CIFAR-10 respectively and best training accuracy of [99.62/99.91].

We then run SLQ with $m = 100$ on the final set of training weights to compute an approximation to the spectrum for both schedules on the full $50,000$ training set (i.e the Empirical Hessian) and with $m = 80$ on an augmented $1,500,000$ data-set as a proxy for the True Hessian, which we denote the *Augmented Hessian*. The 110-Layer ResNet has $1,169,972/1,146,842$ parameters for CIFAR-100/CIFAR-10 respectively, hence for both we have $q < 1$. We primarily comment on the extremal eigenvalues of the Augmented Hessian and their deviation from those of the Empirical Hessian. We also investigate the spectral density at the origin and the change in negative spectral mass. A sharp drop in negative spectral mass, indicates due to Theorem 1 that most of the negative spectral mass is due to the perturbing GOE.

## 6.1 NORMAL SGD SCHEDULE - EMPIRICAL AND AUGMENTED HESSIAN ANALYSIS

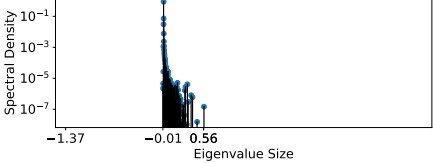

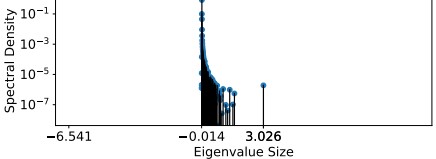

Figure 3: Augmented $1,500,000$ data Hessian spectrum for CIFAR-10 SGD.

Figure 4: Augmented $1,500,000$ data Hessian spectrum for CIFAR-100 SGD.

We plot the Empirical Hessian spectrum for the end of training for CIFAR-100 in Figure 1 and the Augmented Hessian in Figure 4. We plot the same discrete spectral density plots for the Empirical Hessian for CIFAR-10 in Figure 2 and the Augmented Hessian in Figure 3. We note that the effects predicted by Theorem 1 are empirically supported by the differences between the Empirical and Augmented Hessian. For CIFAR-100 the extremal eigenvalues shrink from $[-6.54, 20.56]$ (Figure 1) for the full dataset to $[-0.014, 3.026]$ (Figure 4) for the Augmented dataset. The Empirical Hessian has $43/100$ negative Ritz values with a negative spectral mass total of $0.05$. The Augmented Hessian has $5/80$ negative Ritz values with total negative spectral mass of $0.00016$. For CIFAR-10 the extremal eigenvalues shrink from $[-1.374, 6.821]$ (Figure 2) to $[-0.0061, 0.5825]$ (Figure 3). The fraction of negative Ritz values shrinks from $35/100$ with total weight $0.09$ to $3/80$ with total mass $0.03$.

## 6.2 Overfitting SGD schedule - empirical and augmented Hessian analysis

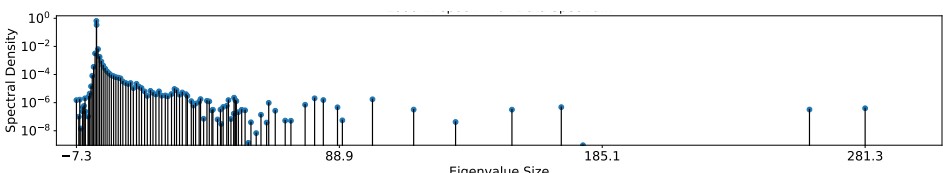

Figure 5: Empirical Hessian Spectrum C100 OverFit SGD.

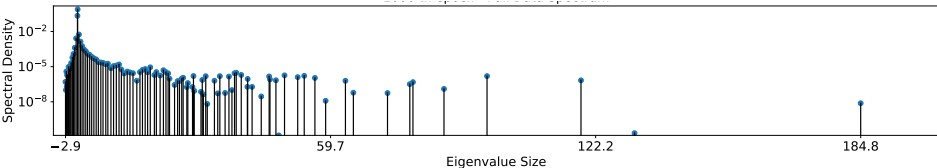

Figure 6: Empirical Hessian Spectrum C10 OverFit SGD.

As can be seen from the spectral plots of the full Hessians for both CIFAR-100 and CIFAR-10 in Figures 6 & 5, and their respective Augmented Hessian 7 & 8, we also observe extreme spectral shrinkage, with the largest eigenvalue decreasing from $[281.32, 184.4]$ to $[45.68, 15.6]$ and the smallest increasing from $[-7.33, -2.9]$ to $[2 \times 10^{-5}, -9 \times 10^{-4}]$ respectively for CIFAR-100/CIFAR-10. We note that despite reasonably extreme spectral shrinkage, the sharpest values of the *Overfit schedule* augmented Hessians are still significantly larger than those of the *Normal schedule*, hence the intuition of sharp minima generalizing poorly still holds for the True Hessian. The fraction of negative Ritz values for the Empirical Hessian is $43/100$, with relative negative spectral mass of $0.032$. For the Augmented Hessian, the number of negative Ritz values is $0$. For the CIFAR-10 Overfit Augmented Hessian, there is one negative Ritz value very close to the origin with a large relative spectral mass $0.086$, hence it could also be part of the spectral peak at the origin.

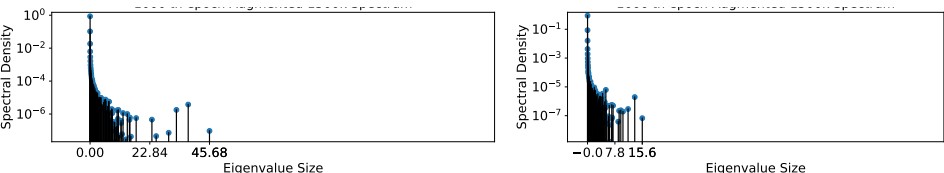

Figure 7: Augmented Data $1,500,000$ Hessian Spectrum C100 OverFit SGD.

Figure 8: Augmented Data $1,500,000$ Hessian Spectrum C10 OverFit SGD.

## 6.3 Rank Degeneracy and Right skew

The vast majority of eigen-directions of the Augmented Hessian are, like those of the Empirical Hessian, locally extremely close to flat. We show this by looking at the 3 Ritz values of largest weight

for all learning rate schedules and datasets, shown in Table 3. The majority of the spectral bulk is carried by the 3 closest weights ($\approx 0.99$) to the origin for all learning rate schedules and datasets. In instances (CIFAR-100 SGD OverFit) where it looks like the largest weight has been reduced, the second largest weight (which is close in spectral distance) is amplified massively.

We further note that, as we move from the $q \gg 1$ regime to $q < 1$, we move from having a symmetric to right skew bulk. Both have a significant spike near the origin. This experimentally corroborates the result derived in (Pennington & Bahri, 2017), where under normality of weights, inputs and further assumptions the symmetric bulk is given by the Wigner semi-circle law and the right skew bulk is given by the Marcenko-Pastur. The huge spectral peak near the origin, indicates that the result lies in a low effective dimension.

For all Augmented spectra, the negative spectral mass shrinks drastically and we see for the *Overfit Schedule* for CIFAR-100 that there is no spectral mass below $0$ and for CIFAR-10 there is one negative Ritz value (and 79 positive). This Ritz value $\lambda_n = -0.00093$ has a weight of $\rho = 0.085$ compared to the largest spike at $\lambda = 0.0001$ with weight $0.889$, hence we cannot rule out that both values belong to a true spectral peak at the origin.

## 7 DISCUSSION

The geometric properties of loss landscapes in deep learning have a profound effect on generalization performance. We introduced the *True Hessian* to investigate the difference between the landscapes for the true and empirical loss surfaces. We derived analytic forms for the perturbation between the extremal eigenvalues of the True and Empirical Hessians, modelling the difference between the two as a Gaussian Orthogonal Ensemble. Moreover, we developed a method for fast eigenvalue computation and visualization, which we used in conjunction with data augmentation to approximate the True Hessian spectrum.

We show both theoretically and empirically that the True Hessian has smaller variation in eigenvalues and that its extremal eigenvalues are smaller in magnitude than the Empirical Hessian. We also show under our framework that we expect the Empirical Hessian to have a greater negative spectral density than the True Hessian and our experiments support this conclusion. This result may provide some insight as to why first order (curvature blind) methods perform so well on neural networks. Reported non-convexity and pathological curvature is far worse for the empirical risk than the true risk, which is what we wish to descend.

The *shape* of the true risk is particularly crucial for understanding how to develop effective procedures for Bayesian deep learning. With a Bayesian approach, we not only want to find a single point that optimizes a risk, but rather to integrate over a loss surface to form a Bayesian model average. The geometric properties of the loss surface, rather than the specific location of optima, therefore greatly influences the predictive distribution in a Bayesian procedure. Furthermore, the posterior representation for neural network weights with popular approaches such as the Laplace approximation has curvature directly defined by the Hessian.

In future work, one could also replace the GOE noise matrix $\varepsilon(\boldsymbol{w})$ with a positive semi-definite white Wishart kernel in order to derive results for the empirical Gauss-Newton and Fisher information matrices, which are by definition positive semi-definite and are commonly employed in second order deep learning (Martens & Grosse, 2015). Our approach to efficient eigenvalue computation and visualization can be used as a general-purpose tool to empirically investigate spectral properties of large matrices in deep learning, such as the Fisher information matrix.

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

# A    TABLE SUMMARIES

| Schedule | $\lambda_1'$ | $\lambda_n'$ | $\lambda_1$ | $\lambda_n$ | Test Acc | Train Acc | Test Loss | Train Loss |
|---|---|---|---|---|---|---|---|---|
| *Normal* | 20.56 | -6.54 | 3.025 | -0.014 | 77.24 | 99.85 | 0.99 | 0.021 |
| *Overfit* | 281.32 | -7.33 | 45.68 | $2 \times 10^{-5}$ | 56.85 | 99.62 | 1.85 | 0.026 |

Table 1: CIFAR-100 statistics table including extremal eigenvalues of the Empirical Hessian $[\lambda_1', \lambda_n']$, the True Hessian $[\lambda_1, \lambda_n]$ and test/train accuracies and losses.

| Schedule | $\lambda_1'$ | $\lambda_n'$ | $\lambda_1$ | $\lambda_n$ | Test Acc | Train Acc | Test Loss | Train Loss |
|---|---|---|---|---|---|---|---|---|
| *Normal* | 6.82 | -1.37 | 3.025 | -0.014 | 95.32 | 99.98 | 0.18 | 0.0034 |
| *Overfit* | 184.4 | -2.9 | 15.6 | $-9 \times 10^{-4}$ | 86.46 | 99.91 | 1.85 | 0.026 |

Table 2: CIFAR-10 statistics table including extremal eigenvalues of the Empirical Hessian $[\lambda_1', \lambda_n']$, the True Hessian $[\lambda_1, \lambda_n]$ and test/train accuracies and losses

| Learning Schedule & DataSet | $[\rho_0, \lambda_0]$ | $[\rho_1, \lambda_1]$ | $[\rho_2, \lambda_2]$ |
|---|---|---|---|
| CIFAR-100 SGD Normal | [0.92,0.0014] | [0.05,-0.03] | [0.018,0.06] |
| CIFAR-100 SGD Normal (Augmented) | [0.84,0.0017] | [0.1,-0.0007] | [0.045,0.003] |
| CIFAR-10 SGD Normal | [0.88,0.007] | [0.085,-0.01] | [0.023,0.02] |
| CIFAR-10 SGD Normal (Augmented) | [0.88,-3.87e-6] | [0.073,0.0006] | [0.03,-0.0007] |
| CIFAR-100 SGD Overfit | [0.65,-0.026] | [0.34,0.05] | [0.006,0.58] |
| CIFAR-100 SGD Overfit (Augmented) | [0.86,1.67e-5] | [0.1,0.0039] | [0.019,0.018] |
| CIFAR-10 SGD Overfit | [0.77,0.01] | [0.22,-0.04] | [0.005,0.32] |
| CIFAR-10 SGD Overfit (Augmented) | [0.89,0.00013] | [0.086,-0.00094] | [0.016,0.0058] |

Table 3: Largest 3 weight Ritz values for different learning schedules and datasets. $[\rho_i, \lambda_i]$ denotes the weight and corresponding eigenvalue $\rho_0 \geq \rho_1 .... \geq \rho_P$.

# B  SGD TRAINING CURVES

Figure 9: SGD CIFAR-100 training and test curves

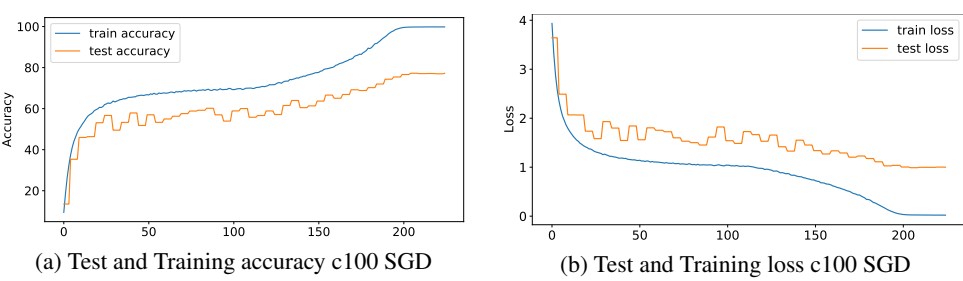

(a) Test and Training accuracy c100 SGD    (b) Test and Training loss c100 SGD

Figure 10: SGD CIFAR-10 training and test curves

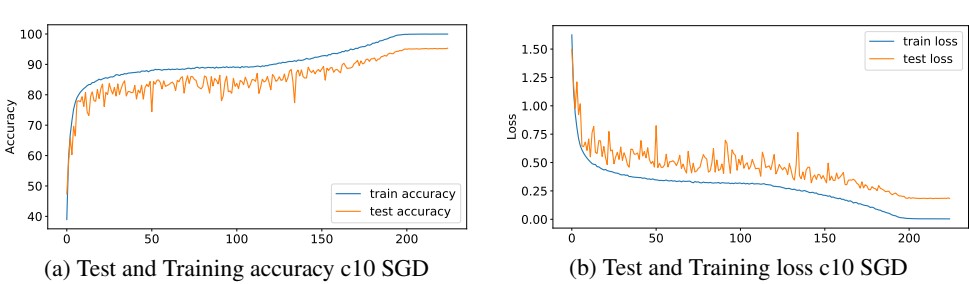

(a) Test and Training accuracy c10 SGD    (b) Test and Training loss c10 SGD

# C  OVERFIT TRAINING CURVES

Figure 11: OverFitSGD CIFAR-100 training and test curves

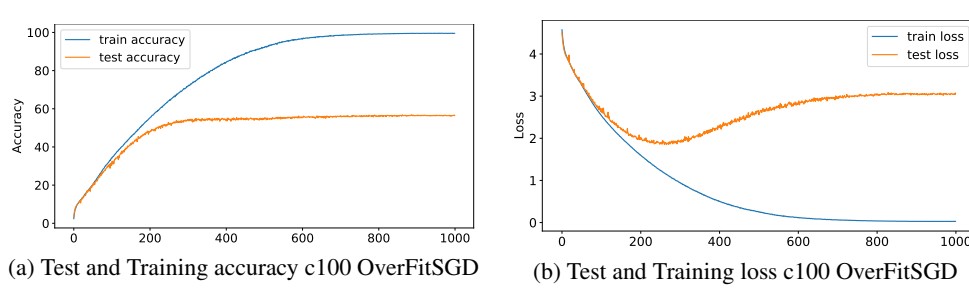

(a) Test and Training accuracy c100 OverFitSGD    (b) Test and Training loss c100 OverFitSGD

Figure 12: OverFitSGD CIFAR-10 training and test curves

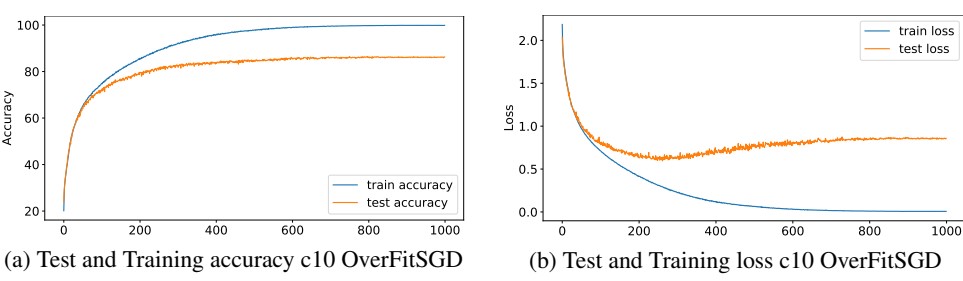

(a) Test and Training accuracy c10 OverFitSGD    (b) Test and Training loss c10 OverFitSGD

## D  RANDOM MATRIX THEORY

Following the notation of (Bun et al., 2017) the resolvent of a matrix $H$ is defined as

$$G_H(z) = (zI_N - H)^{-1} \tag{12}$$

with $z = x + i\eta \in \mathbb{C}$. The normalised trace operator of the resolvent, in the $N \to \infty$ limit

$$\mathcal{S}_N(z) = \frac{1}{N}\text{Tr}[G_H(z)] \xrightarrow{N \to \infty} \mathcal{S}(z) = \int \frac{\rho(u)}{z - u} du \tag{13}$$

is known as the Stieltjes transform of $\rho$. The functional inverse of the Siteltjes transform, is denoted the blue function $\mathcal{B}(\mathcal{S}(z)) = z$. The R transform is defined as

$$\mathcal{R}(w) = \mathcal{B}(w) - \frac{1}{w} \tag{14}$$

crucially for our calculations, it is known that the $\mathcal{R}$ transform of the Wigner ensemble is

$$\mathcal{R}_W(z) = \sigma^2 z \tag{15}$$

**Definition D.1.** *Let $\{Y_i\}$ and $\{Z_{ij}\}_{1 \leq i \leq j}$ be two real-valued families of zero mean, i.i.d random variables, Furthermore suppose that $\mathbb{E}Z_{12}^2 = 1$ and for each $k \in \mathbb{N}$*

$$max(E|Z_{12}^k, E|Y_1|^k) < \infty \tag{16}$$

*Consider an $n \times n$ symettric matrix $M_n$, whose entries are given by*

$$\begin{cases} M_n(i,i) = Y_i \\ M_n(i,j) = Z_{ij} = M_n(j,i), & \text{if } x \geq 1 \end{cases} \tag{17}$$

*The Matrix $M_n$ is known as a real symmetric Wigner matrix.*

**Theorem 2.** *Let $\{M_n\}_{n=1}^{\infty}$ be a sequence of Wigner matrices, and for each $n$ denote $X_n = M_n/\sqrt{n}$. Then $\mu_{X_n}$, converges weakly, almost surely to the semi circle distribution,*

$$\sigma(x)dx = \frac{1}{2\pi}\sqrt{4 - x^2}\mathbf{1}_{|x| \leq 2} \tag{18}$$

the property of freeness for non commutative random matrices can be considered analogously to the moment factorisation property of independent random variables. The normalized trace operator, which is equal to the first moment of the spectral density

$$\psi(H) = \frac{1}{N}\text{Tr}H = \frac{1}{N}\sum_{i=1}^{N}\lambda_i = \int_{\lambda \in \mathcal{D}} d\mu(\lambda)\lambda \tag{19}$$

We say matrices $A\&B$ for which $\psi(A) = \psi(B) = 0^4$ are free if they satisfy for any integers $n_1..n_k$ with $k \in \mathbb{N}^+$

$$\psi(A^{n_1}B^{n_2}A^{n_3}B^{n_4}) = \psi(A^{n_1})\psi(B^{n_2})\psi(A^{n_3})\psi(A^{n_4}) \tag{20}$$

## E  DERIVATION

The Stijeles transform of Wigners semi circle law, can be written as (Tao, 2012)

$$\mathcal{S}_W(z) = \frac{z \pm \sqrt{z^2 - 4\sigma^2}}{2\sigma^2} \tag{21}$$

from the definition of the Blue transform, we hence have

$$z = \frac{\mathcal{B}_W(z) \pm \sqrt{\mathcal{B}_W^2(z) - 4\sigma^2}}{2\sigma^2}$$
$$(2\sigma^2 z - \mathcal{B}_W(z))^2 = \mathcal{B}_W^2(z) - 4\sigma^2 \tag{22}$$
$$\therefore \mathcal{B}_W(z) = \frac{1}{z} + \sigma^2 z$$
$$\therefore \mathcal{R}_W(z) = \sigma^2 z$$

---

[4]We can always consider the transform $A - \psi(A)I$

Computing the $\mathcal{R}$ transform of the rank 1 matrix $H_{true}$, with largest non-trivial eigenvalue $\beta$, on the effect of the spectrum of a matrix $A$, using the Stieltjes transform we easily find following (Bun et al., 2017) that

$$\mathcal{S}_{H_{true}}(u) = \frac{1}{N}\frac{1}{u-\beta} + \left(1 - \frac{1}{N}\right)\frac{1}{u} = \frac{1}{u}\left[1 + \frac{1}{N}\frac{\beta}{1 - u^{-1}\beta}\right] \tag{23}$$

We can use perturbation theory similar to in equation equation 22 to find the blue transform which to leading order gives

$$\mathcal{B}_{H_{true}}(\omega) = \frac{1}{w} + \frac{\beta}{N(1 - \omega\beta)} + \mathcal{O}(N^{-2})$$

$$\mathcal{R}_{H_{true}}(\omega) = \frac{\beta}{N(1 - \omega\beta)} + \mathcal{O}(N^{-2}) \tag{24}$$

setting $\omega = \mathcal{S}_M(z)$

$$z = \mathcal{B}_{H_{true}}(\mathcal{S}_M(z)) + \frac{\beta}{N(1 - \beta\mathcal{S}_M(z))} + \mathcal{O}(N^{-2}) \tag{25}$$

using the ansatz of $\mathcal{S}_M(z) = \mathcal{S}_0(z) + \frac{\mathcal{S}_1(z)}{N} + \mathcal{O}(N^{-2})$ we find that $\mathcal{S}_0(z) = \mathcal{S}_{\epsilon(w)}(z)$ and using that $\mathcal{B}'_M(z) = 1/g'(z)$ , we conclude that

$$\mathcal{S}_1(z) = -\frac{\beta\mathcal{S}'_{\epsilon(w)}(z)}{1 - \mathcal{S}_{\epsilon(w)}(z)\beta} \tag{26}$$

and hence

$$\mathcal{S}_M(z) \approx \mathcal{S}_{\epsilon(w)}(z) - \frac{1}{N}\frac{\beta\mathcal{S}'_{\epsilon(w)}(z)}{1 - \mathcal{S}_{\epsilon(w)}(z)\beta} \tag{27}$$

and hence in the large $N$ limit the correction only survives if $\mathcal{S}_{\epsilon(w)}(z) = 1/\beta$

$$\mathcal{S}_{\epsilon(w)}(z) = \frac{1}{\beta}$$

$$\frac{2\sigma^2}{\beta} = z \pm \sqrt{z^2 - 4\sigma^2} \tag{28}$$

$$\therefore z = \beta + \frac{\sigma^2}{\beta}$$

clearly for $\beta \to -\beta$ we have

$$z = -\beta - \frac{\sigma^2}{\beta} \tag{29}$$

## F   AN INTUITIVE EXPLANATION OF THE KEY RESULT

An extensive linear algebraic and geometric discussion about spectral broadening for sample co-variance matrices can be found in (Ledoit & Wolf, 2004) and whilst the Hessian is not a covariance matrix (the generalized Gauss-Newton or Fisher matrices can be seen as a co-variance of gradients), identical arguments will hold here [5]. The dispersion of the Empirical Hessian eigenvalues around their mean will equal the dispersion of the True Hessian eigenvalues around their mean plus the variance of the Empirical Hessian (from its true value), which is in general $> 0$. Our theoretical assumptions in sections 4.1 and 4.2 allow us to derive analytic results for this broadening. Our assumptions are significantly stricter than those in (Ledoit & Wolf, 2004) and hence automatically fulfil their requirements.

---

[5]the proof of Lemma1 (Ledoit & Wolf, 2004) does not depend on the matrix being PSD

## G   LANCZOS ALGORITHM

In order to empirically analyse properties of modern neural network spectra with tens of millions of parameters $N = \mathcal{O}(10^7)$, we use the Lanczos algorithm (Meurant & Strakoš, 2006) with Hessian vector products using the Pearlmutter trick (Pearlmutter, 1994) with computational cost $\mathcal{O}(NTm)$, where $N$ is the dataset size and $m$ is the number of Lanczos steps. The main properties of the Lanczos algorithm are summarized in the theorems 3,4

**Theorem 3.** *Let $H^{N \times N}$ be a symmetric matrix with eigenvalues $\lambda_1 \geq .. \geq \lambda_n$ and corresponding orthonormal eigenvectors $z_1, ..z_n$. If $\theta_1 \geq .. \geq \theta_m$ are the eigenvalues of the matrix $T_m$ obtained after $m$ Lanczos steps and $q_1, ...q_k$ the corresponding Ritz eigenvectors then*

$$\lambda_1 \geq \theta_1 \geq \lambda_1 - \frac{(\lambda_1 - \lambda_n) \tan^2(\theta_1)}{(c_{k-1}(1 + 2\rho_1))^2}$$

$$\lambda_n \leq \theta_k \leq \lambda_m + \frac{(\lambda_1 - \lambda_n) \tan^2(\theta_1)}{(c_{k-1}(1 + 2\rho_1))^2} \tag{30}$$

*where $c_k$ is the chebyshev polyomial of order $k$*

Proof: see (Golub & Van Loan, 2012).

**Theorem 4.** *The eigenvalues of $T_k$ are the nodes $t_j$ of the Gauss quadrature rule, the weights $w_j$ are the squares of the first elements of the normalized eigenvectors of $T_k$*

Proof: See (Golub & Meurant, 1994). The first term on the RHS of equation 9 using Theorem 4 can be seen as a discrete approximation to the spectral density matching the first $m$ moments $v^T H^m v$ (Golub & Meurant, 1994; Golub & Van Loan, 2012), where $v$ is the initial seed vector. Using the expectation of quadratic forms, for zero mean, unit variance random vectors, using the linearity of trace and expectation

$$\mathbb{E}_v \text{Tr}(v^T H^m v) = \text{Tr}\mathbb{E}_v(vv^T H^m) = \text{Tr}(H^m) = \sum_{i=1}^{N} \lambda_i = N \int_{\lambda \in \mathcal{D}} \lambda d\mu(\lambda) \tag{31}$$

The error between the expectation over the set of all zero mean, unit variance vectors $v$ and the monte carlo sum used in practice can be bounded (Hutchinson, 1990; Roosta-Khorasani & Ascher, 2015). However in the high dimensional regime $N \to \infty$, we expect the squared overlap of each random vector with an eigenvector of $H$, $|v^T \phi_i|^2 \approx \frac{1}{N} \forall i$, with high probability. This result can be seen by computing the moments of the overlap between Rademacher vectors, containing elements $P(v_j = \pm 1) = 0.5$. Further analytical results for Gaussian vectors have been obtained (Cai et al., 2013).

## H   SPECTRAL PLOTS - CIFAR-100 PRERESNET

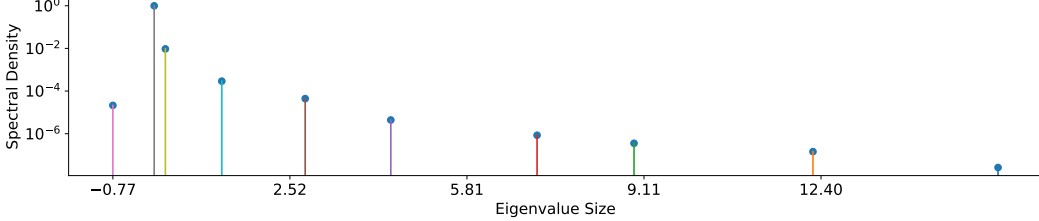

Figure 13: Augmented CIFAR-100 PreResNet 110 Eigenspectrum Epoch 300, 5 million samples and $m = 10$ Lanczos steps

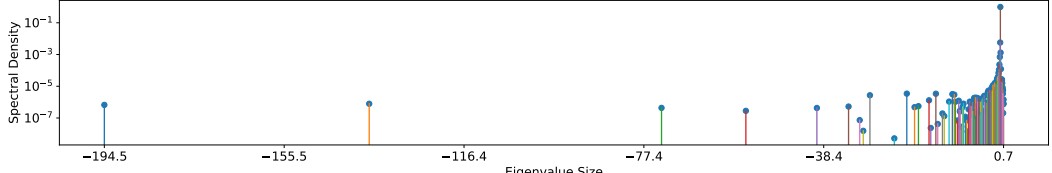

Figure 14: CIFAR-100 PreResNet110 0'th epoch full dataspectrum

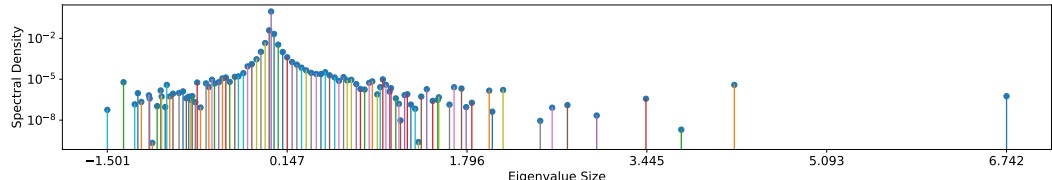

Figure 15: CIFAR-100 PreResNet110 25'th epoch full dataspectrum

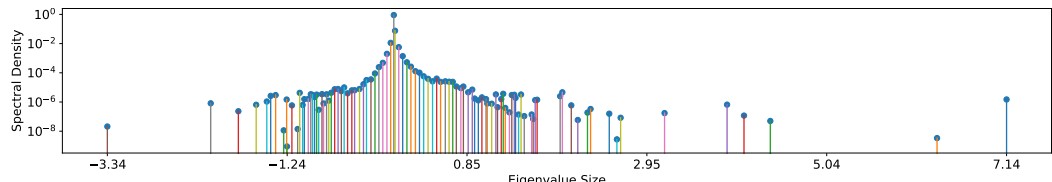

Figure 16: CIFAR-100 PreResNet110 50'th epoch full dataspectrum

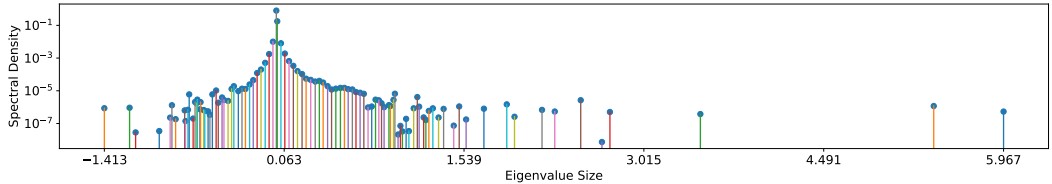

Figure 17: CIFAR-100 PreResNet110 75'th epoch full dataspectrum

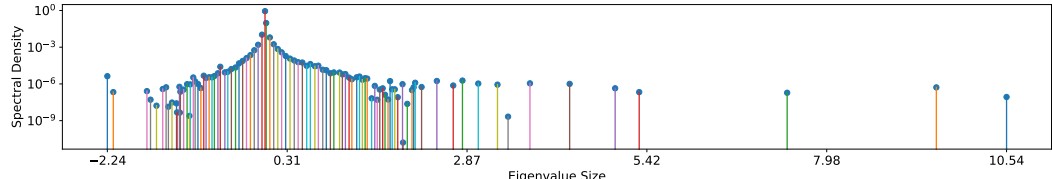

Figure 18: CIFAR-100 PreResNet110 125'th epoch full dataspectrum

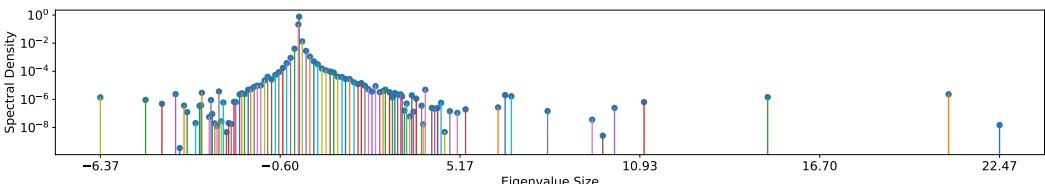

Figure 19: CIFAR-100 PreResNet110 150'th epoch full dataspectrum

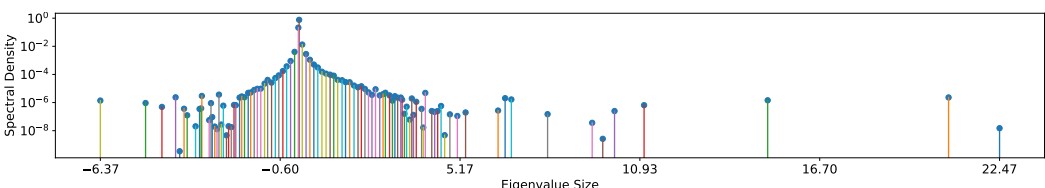

Figure 20: CIFAR-100 PreResNet110 150'th epoch full dataspectrum

# I SPECTRAL PLOTS - CIFAR-100 VGG16BN

The number of Parameters in the $VGG - 16$ network is $P = 15291300$ so our augmentation procedure is unable to probe the limit $q < 1$. The training procedure is identical to the PreResNet except for an initial learning rate of $0.05$ and $T = 300$ epochs. Here we also see a reduction in both extremal eigenvalues.

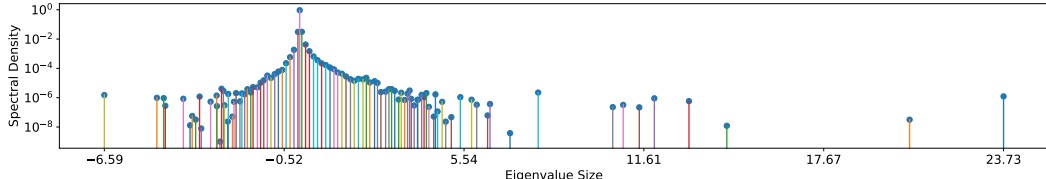

Figure 21: CIFAR-100 VGG16BN 300'th epoch full dataspectrum

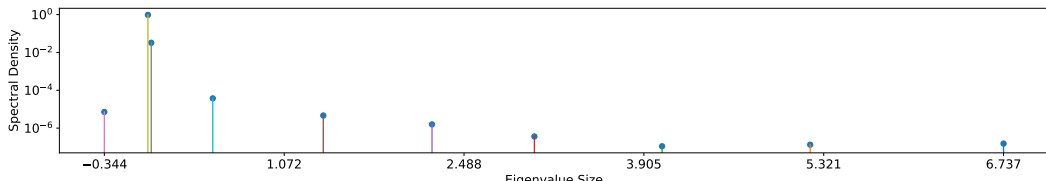

Figure 22: Augmented CIFAR-100 VGG16BN Eigenspectrum Epoch 300, 5 million samples and $m = 10$ Lanczos steps

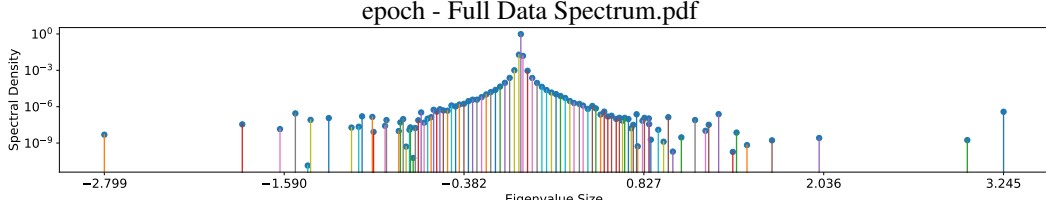

Figure 23: CIFAR-100 VGG16BN 0'th epoch full dataspectrum

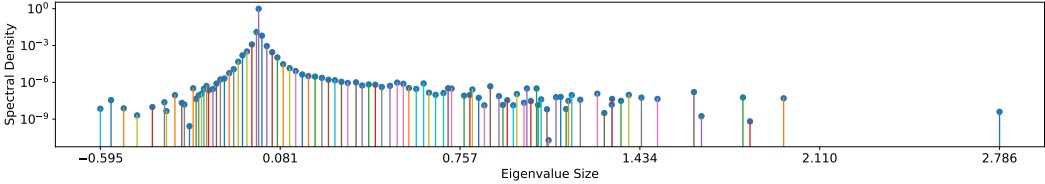

Figure 24: CIFAR-100 VGG16BN 25'th epoch full dataspectrum

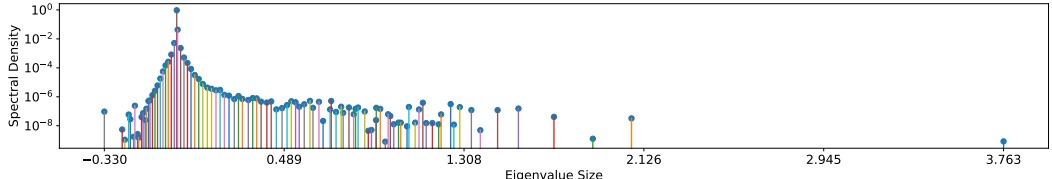

Figure 25: CIFAR-100 VGG16BN 50'th epoch full dataspectrum

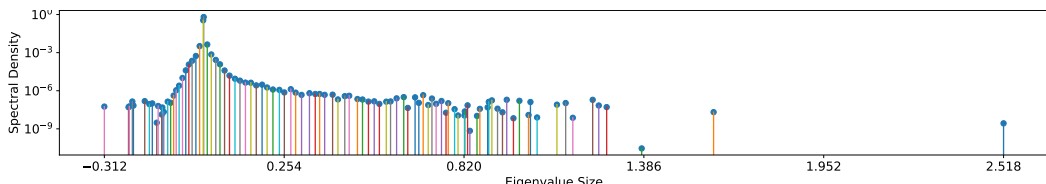

Figure 26: CIFAR-100 VGG16BN 75'th epoch full dataspectrum

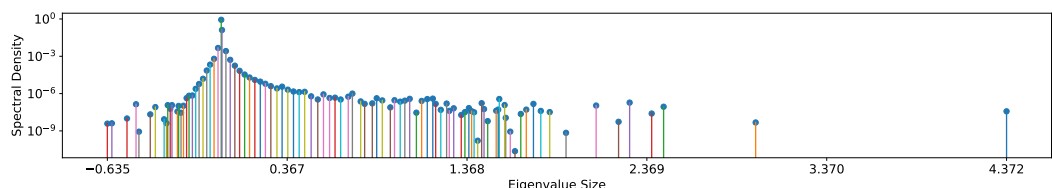

Figure 27: CIFAR-100 VGG16BN 125'th epoch full dataspectrum

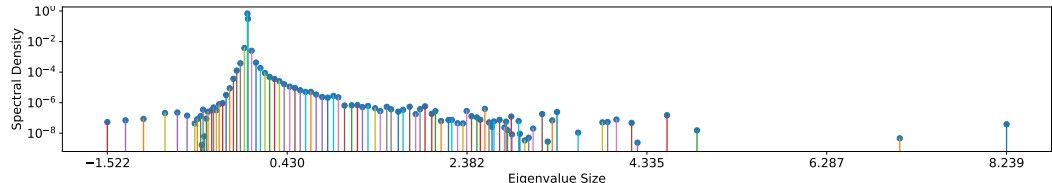

Figure 28: CIFAR-100 VGG16BN 150'th epoch full dataspectrum

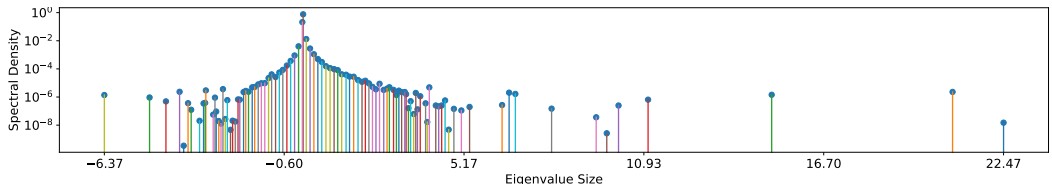

Figure 29: CIFAR-100 VGG16BN 150'th epoch full dataspectrum

# J   SPECTRAL PLOTS - CIFAR-10 PRERESNET

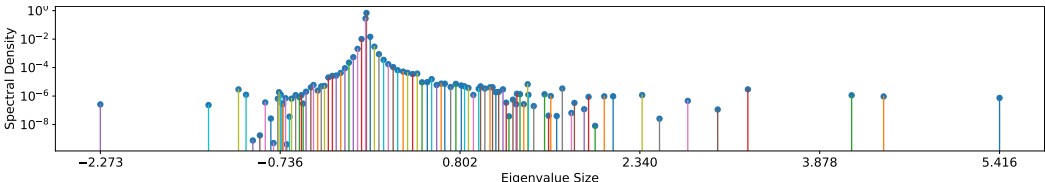

Figure 30: CIFAR-10 PreResNet110 50'th epoch full dataspectrum

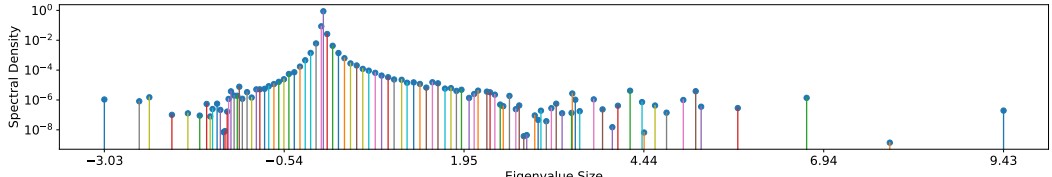

Figure 31: CIFAR-10 PreResNet110 300'th epoch full dataspectrum

# K  SPECTRAL PLOTS - CIFAR-10 VGG16BN

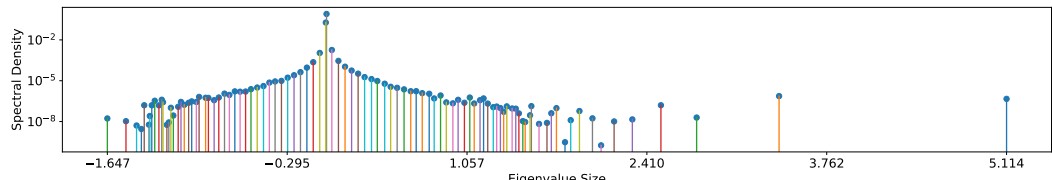

Figure 32: CIFAR-10 VGG16BN 0'th epoch full dataspectrum

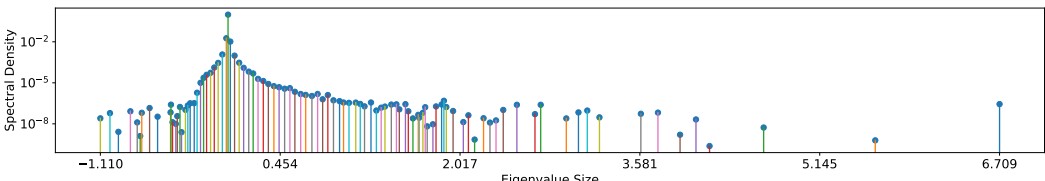

Figure 33: CIFAR-10 VGG16BN 300'th epoch full dataspectrum

