# OpenReview forum: "Towards understanding the true loss surface of deep neural networks using random matrix theory and iterative spectral methods"
_ICLR.cc/2020/Conference — Reject_

### Official Review · AnonReviewer3 · 2019-10-18
**Official Blind Review #3**

**Rating:** 3

**Review:**

This paper analyzes the spectrum of the Hessian matrix of large neural networks, both from the theoretical and the empirical perspective. The main contributions are (1) a theoretical analysis of the max/min eigenvalues, showing that the max/min are larger/smaller for empirical Hessians compared to "true" Hessians, and (2) an empirical analysis and visualization of spectra using a Lanczos quadrature approach for a variety of training methods and architectures.

Generally, analyzing the local curvature of (approximate) optima is an important problem for building a better understanding of neural network loss surfaces and for shedding light on how that geometry might be related to generalization performance. Therefore, I find the topic of this paper salient, and the questions analyzed are sure to be of interest to a broad range of theoretically-minded and practically-minded deep learning researchers.

However, I did not find the theoretical analysis to be especially novel, deep, or informative. I also did not find the conclusions reached to be particularly compelling, especially insofar as they related to how the geometry might affect generalization performance. For these reasons, I think this paper is a bit below the bar for acceptance, and I'd encourage the authors to add more content and depth to their analysis in a future submission.

Let me elaborate on my above comments in slightly more detail.


On the Theoretical analysis:

The general observation being made here is that adding more data will tighten the spectrum. This behavior is well-known, e.g. simply adjusting the aspect ratio of the Marcenko-Pastur distribution gives the same effect. Similar behavior is known in more complicated models. The result in Theorem 1 is not especially novel and is probably available in the literature in some form (I didn't find an immediate reference, but even without one, and even if the precise result isn't presented formally in prior work, this type of result is quite well-known).

Some of the assumptions are very strong, especially the independence of the entries in \epsilon. While I actually suspect that the overall conclusion of spectral broadening may be fairly robust, the argument in section 4.3 did not convince me. There could be very strong correlations in \epsilon that get induced by the architecture and learned weights, and these correlations could become larger as the dataset size grows, leading to big discrepancies with the predictions. This would probably be most pronounced when transitioning from the under-parameterized to over-parameterized regimes.


On the Empirical analysis:

As the authors note, the main Lanczos quadrature method being utilized was introduced in several prior works, and the "improvement" offered here is basically a small tweak (reducing the number of random vectors). The argument in section 5.2 for reducing this number to one was not particularly compelling, and if the author's regarded this modification as an important contribution, I would have appreciated a more thorough variance analysis, both from the theoretical and empirical perspectives.

Additionally, related to error analysis, the method utilized in this work (and in the prior works that this work build upon) does not really have a robust way of reliably measuring the error in the spectral density estimation. There is no guarantee that the estimate will be accurate. Such an error estimate should depend on the ground-truth spectrum, and if the ground-truth spectrum is particularly pathological (say, having a large density near zero and a sizable number of large outliers  [which may in fact be the case that may happen in practice!]), the estimate may be quite bad. So I'm actually not completely convinced that the results presented here are accurate estimates of the actual spectrum.

Finally, I found the method for generating the "True Hessian" to be quite ad-hoc and likely severely biased. To what extent are the results skewed by the particular data augmentation procedure being utilized? Any effort to control for this kind of bias would be an important addition to this paper. Additionally, it might have been useful to corroborate some of the main claims with regard to the True Hessian (vs Empirical Hessian) by utilizing a synthetic data distribution from which unbiased samples could be drawn. Lastly, even though the number of augmented samples was large, it was not very large compared to the number of parameters. Perhaps studying a regime in which the ratio was farther from one would have been useful.


On the conclusions:

One of the main important motivations for this analysis is to understand how curvature might relate to generalization. This motivation was put forward in the introduction and was the basis for the experiments comparing the Normal and Overfit model classes. However, I found this single type of comparison to provide very weak evidence in favor of the conclusion that flat minima generalize better. A more compelling argument in this direction would have included substantially more empirical evidence, or (perhaps in better keeping with the main perspectives of this paper) additional theoretical analysis of this particular point.

_____________

Overall, I think the problem studied in this paper is quite interesting, but I did not find the theoretical or empirical contributions to be sufficiently novel, deep, or informative to merit publication in their present form.

**Experience Assessment:**

I have published in this field for several years.

**Review Assessment: Checking Correctness Of Derivations And Theory:**

I assessed the sensibility of the derivations and theory.

**Review Assessment: Checking Correctness Of Experiments:**

I assessed the sensibility of the experiments.

**Review Assessment: Thoroughness In Paper Reading:**

I read the paper thoroughly.

---

> ### Author Response · Authors · 2019-11-15
> **The theory including scaling as is relevant to deep learning is not known, Lanczos is significantly more accurate a spectral estimation tool than what is commonly used and the results for sampling from explicit distributions are known.**
>
> Thank you for your thorough response.
>
> With random matrix theory there is a natural link between the neural network Hessian and covariance matrices (which is not strict due to the fact that the Hessian is non PSD) it is possible to use the aspect ratio of the Marcenko pastur to predict spectral broadening, although the final form is quantitatively different to what we have derived. Given that the Hessian is not a covariance matrix (it is not even a covariance of gradients), with negative eigenvalues, we make exactly this intuition rigorous from the ground up and believe this to be a valuable contribution. We explicitly formulate a potential noise model for the true Hessian, map that onto the GOE and then derive the full result using scaling (which we have not seen elsewhere in the literature) and make that presentable to the deep learning community. We have not seen a discussion on the true risk surface, or any theoretical results on the eigenvalues wrt to the empirical risk surface and we see this as novel and compelling.
>
> Under what conditions do you predict the noise per element of the Hessian having strong correlations that are of O(P^{2})?
>
> Bounds on the accuracy of the Ritz values are given in (‘Meurant’, ‘Lanczos’) and are well known. Furthermore, the spectrum given is a discrete moment matched approximation to the underlying spectrum ( in our case 100 moments), which is significantly more accurate than other works using power iteration (a less optimal algorithm than Lanczos), or the diagonal of the Fisher information or Hessian. Could you please reference or at least explain why the estimate of a sizeable number of outliers with Lanczos would be bad? Or having a large density near zero? Our spectral density estimation plot captures both these features and we are unaware of any literature that in any way indicates the inability for Lanczos to deal with these “pathologies”
>
> Extensive simulations on the broadening effect of variables sampled from known distributions has been documented in the literature (J.Bun – ‘Cleaning large correlation matrices: tools from random matrix theory’), which is rather uninformative to deep learning practitioners or people interested in the effect of commonly used augmentation procedures on the spectrum and process. For the case of Linear regression, the true Hessian is just the covariance matrix under the data generating distribution, for which extensive results can be found in the above reference.

---

### Official Review · AnonReviewer1 · 2019-10-21
**Official Blind Review #1**

**Rating:** 3

**Review:**


In this paper, the authors uses the random matrix theory to study the spectrum distribution of the empirical Hessian and true Hessian for deep learning, and proposed an efficient spectrum visualization methods. The results obtained in the paper can shed some lights on the understanding of existing optimization algorithms (e.g., first order methods).

While this paper is quite interesting, I kind of feel that it has limited value to the research community due to the following concerns.

1)	The work is based on some assumptions, which is however not very reasonable.  Assumptions 1 and 2 are too strong – it is difficult to know and to guarantee that the elements of \epsilon are i.i.d. Gaussian distributed, and it is not guaranteed that the true Hessian is of low rank. I feel these assumptions are key to the proof, but they make the impact of the work lowers on practical situations where we have no knowledge and guarantee on the true Hessian and its relationship with the empirical Hessian. Although the authors made some discussions on generalizing the assumptions, it is unclear from the limited discussions whether the same theoretical results could be obtained in the generalized setting.

2)	The experimental verification of the theoretical results is not indirect and somehow not very convincing. Because the true Hessian is hard to obtain, the authors compared the empirical Hessian with respect to different sizes of the training data as a proxy. However, this is not convincing since we do not know whether the same results can still be observed if further increasing the data scale, and which is the trend with respect to the increasing scale. A better way would be to design a simulation experiments, in which we know the data distribution (and thus can compute the true Hessian).

3)	 The experiments were only done on the CIFAR datasets – not at large scale and not for diverse applications. So it is hard to generalize the experimental results to the entire space of “deep learning” (as indicated by the title).

4)	The practical value of the work in not very clear to me. The authors only made limited discussions on this – it can be used to explain why first-order optimization algorithms work well in practice. However, this is what we already know. It would be much better if some practical guidelines can be derived which can improve (but not just explain) the optimization process of deep learning.

**I read the author response, however, I do not think I am convinced to change my rating.

**Experience Assessment:**

I have published in this field for several years.

**Review Assessment: Checking Correctness Of Derivations And Theory:**

I assessed the sensibility of the derivations and theory.

**Review Assessment: Checking Correctness Of Experiments:**

I carefully checked the experiments.

**Review Assessment: Thoroughness In Paper Reading:**

I read the paper at least twice and used my best judgement in assessing the paper.

---

> ### Author Response · Authors · 2019-11-15
> **deep learning spectral theory papers to date employ significantly more assumptions than we do, the broadening on known distributions is well documented,  the theory is not specific to CIFAR and the true risk surface is relevant to traversing and minimizing the true risk**
>
> Deep learning spectral theory papers to date (e.g. [Choromanska – ‘The Loss Surfaces of Multilayer Networks’], [Pennington – ‘Geometry of Neural Network Loss Surfaces via Random Matrix Theory’]) assume a combination of independence of inputs, independence of weights, path independence, residual independence, independence of Hessian elements and free addition. This is the first paper in the field that reduces the number of assumptions, to the point that we allow arbitrary dependency. The noise matrix is i.i.d, quite a standard assumption in probabilistic analysis. We have written in section 4.3 how these assumptions could be relaxed in return for other more technical conditions. A certain level of statistical independence is necessitated for extensions of the central limit theorem to hold (Stein – ‘A bound for the error in the normal approximation to the distribution of a sum of dependent random variables’) and there currently exists no mathematics to deal with the general fully dependent problem. Could the reviewer please provide an example of spectral deep learning theory where a smaller set of assumptions have been made?
>
> Extensive simulations on the broadening effect of variables sampled from known distributions has been documented in the literature (J.Bun – ‘Cleaning large correlation matrices: tools from random matrix theory’), which is rather uninformative to deep learning practitioners or people interested in the effect of commonly used augmentation procedures on the spectrum and process. For the case of Linear regression, the true Hessian is just the covariance matrix under the data generating distribution, for which extensive results can be found in the above reference. Our horizontal flips and crops (as detailed in section 6) bring into focus the finite size effects on the spectrum for a real network on a real dataset. As a gedanken-experiment, if the crops and flips were so different from new draws of the data generating distribution, why does training with these procedures result in significantly improved test accuracy? (which can be seen as independent draws from the generating distribution).
>
> None of the theory is CIFAR specific, we just exemplify the result on commonly used datasets and architecture as opposed to the usual toy examples on two layer fully connected networks. Which datasets would you like to see us consider to significantly improve your score?
>
> Optimization is generally concerned with minimizing the empirical risk. The true risk, which can be seen as preferred objective for learning has been considered completely intractable to study. In this work we specifically study very general properties of the true risk surface. Clearly knowing how to traverse and minimise the true risk would remove the generalisation gap and allow us to perform strictly better and is of high value. Knowing how its curvature properties vary from that of the empirical risk could be instrumental in this task.

---

### Official Review · AnonReviewer2 · 2019-10-22
**Official Blind Review #2**

**Rating:** 3

**Review:**

The paper compares the true hessian and the empirical hessian of the loss function, showing the spectrum of the Empirical Hessian is generally broadened, and proposes a way to visualize the spectrum. The paper is well written with sound theoretical reasoning and empirical validation. However, the reviewer has the following concerns:

As to the assumption in this paper, it is too strong to assume independence between different elements of Hessian, Wigner ensemble seems to be a better model, since the only independence only comes from samples instead of the elements of the Hessian. This would require a completely different analysis to deal with.
Furthermore, it seems that Lemma 1 and Theorem 1 are asymptotic results. I don’t see how these are related to the empirical validation, since in experiments the compared dataset are CIFAR-10/CIFAR-100 and augmented CIFAR-10/CIFAR-100, which only differs in the number of samples. It makes more sense to have a non-asymptotic analysis here.


**Experience Assessment:**

I have read many papers in this area.

**Review Assessment: Checking Correctness Of Derivations And Theory:**

I assessed the sensibility of the derivations and theory.

**Review Assessment: Checking Correctness Of Experiments:**

I assessed the sensibility of the experiments.

**Review Assessment: Thoroughness In Paper Reading:**

I read the paper at least twice and used my best judgement in assessing the paper.

---

> ### Author Response · Authors · 2019-11-15
> **Our Hessian model is a Wigner matrix + low rank structure. Lemma 1 has no asymptotics.**
>
> Dear reviewer 2, thank you for your compliments on the writing and theory of our paper.
>
> Compared to other works our model accounts for structure and allows for dependence in the Hessian elements. We model the empirical Hessian as low-rank matrix ( the source of the outlier eigenvalues corresponding to structure which are extensively empirically observed) + noise. Existing works assume that each element has its own distribution (independent from other elements) and do not account for the structure.
>
> The Wigner ensemble explicitly requires elements to be independent (up to the Hermitian condition) and this is in fact the exact noise model we use for the noise matrix (the GOE is a special case of the Wigner ensemble which further possesses rotational invariance). We would be grateful if you could  clarify further your criticisms and what would need to be performed to significantly improve the score. For example, what “completely different analysis” do we need? We already model the structure of the true Hessian, which can have arbitrarily large dependence structure, as a perturbation of a Wigner ensemble, which from your review seemed to be your ideal noise model?
>
> Lemma 1 has no asymptotics, and Theorem 1 is in the asymptotic regime where the number of samples of P and N go to infinity. The central limit theorem is also only valid asymptotically (finite P and finite N), furthermore we have explicitly mentioned how the finite order corrections can be derived in section 4.2. Specifically that they drop off as P^{-2/5} where for Neural networks P is large, usually tens of millions.
>
> The empirical validation focuses on a simple question, is the true risk surface smoother (smaller spectral norm or maximal eigenvalues) and that of the empirical risk surface? If we had an infinite number of samples the empirical risk surface would converge (central limit theorem) to the true risk surface, hence changing the number of samples and the effect on the spectrum is exactly what we are trying to probe in our hypothesis.

---

### Decision · Program_Chairs · 2019-12-19

**Decision:**

Reject

**Comment:**

The reviewers all appreciated the importance of the topic: understanding the local geometry of loss surfaces of large models is viewed as critical to understand generalization and design better optimization methods.

However, reviewers also pointed out the strength of the assumptions and the limitations of the empirical study. Despite the claim that these assumptions are weaker than those made in prior work, this did not convince the reviewers that the conclusion could be applied to common loss landscapes.

I encourage the authors to address the points made by the reviewers and submit an updated version to a later conference.